# Evolution of intraocular pressure after cataract surgery in nonglaucomatous patients: A post-hoc analysis of PERCEPOLIS clinical trial data

Jean-Marc Perone[1]*, Jean-Charles Vermion[1], Yinka Zevering[2], Julie Francois[1], Alice Nesseler[1], Grace Gan[1], Christophe Goetz[2]

1 Department of Ophthalmology, Mercy Hospital, Metz-Thionville Regional Hospital Center, Metz, Grand-Est, France, 2 Clinical Research Support Unit, Mercy Hospital, Metz-Thionville Regional Hospital Center, Metz, Grand-Est, France

* jm.perone@chr-metz-thionville.fr

## Abstract

### Purpose

Intraocular pressure (IOP) drops after cataract surgery, including in nonglaucomatous eyes, but the mechanisms and factors determining this change remain unclear. Thus, post-hoc analysis was conducted on PERCEPOLIS, a large randomized-controlled trial showing non-inferior endothelial-cell loss between two cataract-extraction techniques in nonglaucomatous eyes.

### Methods

Eyes with preoperative and 1-, 3-, and 12-month IOP measurements were identified. None received IOP-lowering medication before or after surgery. To identify possible mechanisms driving postoperative IOP drop, the cohort was categorized according to preoperative IOP (High: 20–29; Intermediate: 15–19; Low: 10–14 mmHg). IOP change in the whole cohort and three subgroups was determined. Multivariable analysis assessed whether age, sex, cataract density, preoperative IOP, anterior-chamber depth, axial length, lens thickness, lens position, relative lens position, cataract-extraction method, effective phaco time, or implant power predicted IOP drop at 3 months in the whole cohort and each subgroup. The literature was reviewed to determine the influence of regression-to-the-mean, a misleading statistical phenomenon, on postoperative IOP drop.

### Results

Whole-cohort IOP dropped from 17.6±3.5 mmHg by 1.9±0.2 (1 month), 2.4±0.2 (3 months), and 1.7±0.2 (12 months) mmHg. IOP drop was greatest in the High subgroup. Preoperative IOP predicted 3-month IOP change in the whole cohort (beta=0.53±0.04, $p<0.001$) and all subgroups (range: beta=0.36–0.72±0.15–0.30,

**Data availability statement:** The datasets generated during and/or analyzed during the current study are not publicly available according to French Law No. 2018-493 of June 20, 2018 on the protection of personal data (The General Data Protection Regulation (Regulation (EU) 2016/679) (GDPR: article 9) but are available from the Clinical Research Support Platform (Plateforme d'Appui à la Recherche Clinique [PARC]) of the Regional Central Hospital (CHR) of Metz-Thionville on reasonable request (email: projetrechercheclinique@chr-metz-thionville.fr, tel: +33 3 87 17 98 82). All non-archived data is subject to daily backups while all archived data is subject to duplicate storage at two different sites. This data processing is compliant with a baseline reference methodology (MR001) to which the CHR MetzThionville signed a compliance commitment on October 8, 2018.

**Funding:** The author(s) received no specific funding for this work.

**Competing interests:** The authors have declared that no competing interests exist.

$p = 0.0505–0.001$). Male sex also predicted larger whole-cohort IOP drop (beta = 0.79, $p = 0.01$). Different variables predicted greater IOP change in the three subgroups: male sex in High (beta = 1.63 ± 0.57, $p = 0.005$), higher implant power in Intermediate (beta = 0.19 ± 0.07, $p = 0.01$), and hard cataract in Low (beta = 2.14 ± 0.89, $p = 0.02$). Literature review suggested that regression-to-the-mean accounts for only a small proportion of IOP drop after cataract surgery.

## Conclusions

The IOP drop after cataract surgery may largely be a real biological effect. In normotensive eyes, the strongest predictor of postoperative IOP drop was preoperative IOP. Distinct mechanisms may mediate the IOP change in normotensive eyes.

## Introduction

Cataracts were a leading cause of blindness in industrialized countries until a few decades ago, when improvements in cataract surgery started leading to excellent visual outcomes with few serious complications [1]. Cataract surgery has consequently become the most common surgical procedure in the world [2].

Another leading global cause of vision loss is glaucoma, which is a group of optic neuropathies characterized by progressive degeneration of retinal ganglion cells that results in cupping of the optic disc. The most common cause is excessive intraocular pressure (IOP), which is generally induced by imbalances between the secretion of aqueous humor by the ciliary body and its drainage by the trabecular meshwork and uveoscleral outflow pathways. Primary glaucoma is broadly categorized according to the type of outflow pathway blockage. Thus, in primary angle-closure glaucoma (PACG), the iris is squeezed against the cornea, which compresses and blocks the trabecular meshwork, Schlemm's canal, and sometimes also the uveoscleral drains. It also narrows or occludes the iris-lens canal, hampering the circulation of aqueous humor from the posterior to the anterior chamber. By contrast, in primary open-angle glaucoma (POAG), the iris is positioned correctly but the trabecular meshwork does not drain properly [3].

Since older age is a risk factor for both cataracts and glaucoma, the conditions frequently co-exist: in 2006, 30% of patients who underwent cataract surgery in the US had concomitant glaucoma [4]. Starting in the 1970s, it was noted that the IOP of glaucoma patients drops after cataract surgery [5]. Multiple studies and meta-analyses have since confirmed this [4,6–12]. Notably, the phenomenon has also been observed repeatedly in nonglaucomatous patients (S1 Table, S1 Fig) [13–35]. In PACG cases, the post-cataract surgery drop in IOP is probably largely due to the fact that the surgery widens the iridocorneal angle, thus relieving the blockage of the trabecular meshwork and iris-lens canal. By contrast, in POAG cases, it has been suggested that remodeling of the trabecular meshwork explains the IOP drop. However, the mechanisms that drive the improvement in IOP in nonglaucomatous cases remain unclear [36]. To clarify these mechanisms, many studies have searched for preoperative, perioperative, and postoperative factors that might drive the IOP drop

after cataract surgery in glaucomatous and nonglaucomatous patients. However, most of these studies were retrospective, involved small sample sizes, examined only a few variables, conducted univariable analyses only, and/or did not exclude the use of IOP-lowering medications after surgery (S2 Table) [13–15,17,18,20,23–25,28,29,32–35,37–58].

PERCEPOLIS is a randomized controlled trial (RCT) that compared two phacoemulsification techniques, namely, subluxation and divide-and-conquer (DAC), in terms of postoperative endothelial-cell loss at 1, 3, and 12 months. Subluxation is a supracapsular technique where the nucleus is transposed into the anterior chamber and then fragmented [59,60]. DAC is an endocapsular method where the nucleus is fractured into four quadrants in the capsular bag and then removed [61]. The study cohort consisted of 292 nonglaucomatous patients and IOP was measured before and after surgery with calibrated instruments according to strict guidelines. The trial showed that subluxation is non-inferior to DAC with regard to endothelial-cell loss [62]. Notably, the trial not only involved a large sample size and rigorously acquired trial data, all trial patients were naïve for IOP-lowering medication before and after surgery. These features led us to use PERCEPOLIS data to address the post-hoc question, which of 12 patient/eye and surgical factors can predict the IOP drop in nonglaucomatous eyes after cataract surgery? Since preoperative IOP was found to be the strongest predictor of IOP drop, we also explored whether the variables that predicted IOP drop differed in patients with high, intermediate, and low preoperative IOP. We discuss these results and whether the IOP drop in nonglaucomatous and glaucomatous patients is a real biological response or merely reflects a misleading statistical phenomenon known as regression-to-the-mean (RTM).

## Methods

### Study design and ethics

PERCEPOLIS (PERte Cellulaire Endotheliale après PhacOemuLsification Intra ou Supracapsulaire/endothelial cell loss after supra- or endocapsular phacoemulsification) is a single-center parallel-arm randomized non-inferiority clinical trial (ClinicalTrials.gov Identifier: NCT02535819, IDRCB 2015-A00789-49). Patients were recruited on 4 June 2015–4 April 2016 at the Metz-Thionville Regional Hospital Center (Metz, France). The study was approved by the Ethics Committee of the French Society of Ophthalmology (IRB 00008855 Société Française d'Ophtalmologie IRB#1) and adhered to the tenets of the Declaration of Helsinki. All subjects provided informed written consent before randomization. JMP had access to information that could identify individual participants during or after data collection.

### PERCEPOLIS trial details

The PERCEPOLIS trial has been described previously [62]. The cohort was a convenience series of 292 adult (≥18 years) patients who (i) had a nuclear (NO1–NO4; NC1–NC4), cortical (C1–C5), or subcapsular (P1–P5) cataract of normal–severe density, as determined by LOCSIII classification [63], (ii) had best spectacle-corrected visual acuity (BSCVA) of>+0.2 logMAR, and (iii) underwent subluxation ($n = 148$) or DAC ($n = 144$) on one eye. Exclusion criteria included glaucoma, as indicated by a history of glaucoma, chronic glaucoma treatment, or abnormalities of the optic nerves and ganglion fibers on preoperative optical coherence tomography. Other exclusion criteria were: white/brown cataracts; insulin-dependent diabetes or diabetic retinopathy; preexisting corneal, ocular tone, or posterior segment pathology; preoperative endothelial-cell density <1500 cells/mm$^2$; pregnancy; and history of retinal detachment, ocular trauma, or anterior/posterior segment surgery. Patients undergoing additional procedures apart from cataract removal and intraocular-lens implantation were also excluded. The randomization and single-blinding procedures used in the RCT have been described in detail previously [62]. The patients were followed up 1, 3, and 12 months after surgery by outpatient visits.

### Phacoemulsification surgery

The vast majority of surgeries (88%) in the RCT were performed by one experienced surgeon (JMP, the principal investigator of the RCT). The procedure was as described [62]. Briefly, after topical anesthesia with 1.8-mg/0.4-mL

oxyprocaine chlorhydrate (Laboratoires Théa, France), pupillary dilation was induced with 0.28/5.4-mg tropicamide/phenylephrine hydrochloride (Mydriasert; Laboratoires Théa, France). A blepharostat was then positioned, a coaxial 2.2-mm corneal mini-incision was generated, dispersive viscoelastic (DuoVisc; Alcon Laboratories, Switzerland) was injected, a second incision was created with a 20-gauge needle, circular capsulorhexis was performed, and lens-nucleus hydro-dissection was conducted with either the DAC [61] or the subluxation (Garde-à-vous) [59,60] technique with a Stellaris (year 2015; Bausch & Lomb, Inc., Vaughan, Ontario, Canada). Surgery was completed with a corneal stitch if required and patients were given an intracameral injection of cefuroxime (Aprokam; Laboratoires Théa, France).

### Pre- and postoperative variables

All IOP measurements were conducted in the same way [64] with a slit lamp-calibrated Goldmann applanation tonometer (BQ 900; year 2012; Haag-Streit, Koniz, Switzerland). Specifically, at each study visit, IOP was measured three times in immediate succession between 10:00 and 12:00. The measurement average was then recorded.

Along with preoperative and postoperative IOP, the following variables were collected: patient age, sex, and cataract density; preoperative anterior-chamber depth (ACD), axial length (AXL), and lens thickness (LT) (measured with IOL Master 700; year 2015, Carl Seiss Meditec AG, Jena, Germany); and type of surgery, effective phaco time (EPT), and implant power. Age, sex, ocular-anatomical variables, and EPT were chosen because previous studies suggested that they might affect postoperative IOP (S2 Table) [13–15,17,18,20,23–25,28,29,32–35,37–58]. EPT is the time in seconds where the ultrasound is at 100% power (calculated as phaco time × average phaco power). Two composite preoperative ocular-anatomical variables that are based on ACD, AXL, and LT were also analyzed because some studies suggest that they can associate with or predict IOP reduction after cataract surgery, including in nonglaucomatous eyes (S3 Table) [14,15,17,23,29,32–35,37,39,41–49,51–53,55,57,58,65–77]: (i) lens position (LP), which is ACD + 0.5LT; and (ii) relative LP (RLP), which is LP/AXL. Smaller LP and RLP values signify a more anteriorly positioned lens [45].

### Post-hoc study inclusion criteria and endpoints

The present study focused on the PERCEPOLIS patients in whom IOP was measured before cataract surgery and at one or more timepoints (1, 3, and/or 12 months) after surgery. To minimize possible confounding due to differences in surgeon experience, we excluded eyes whose surgery had been conducted by surgeons other than the principal investigator (JMP). The 3-month visit group had the most patients/eyes (n = 241) and therefore constituted the 'whole cohort'. Of these patients/eyes, 238 (99%) and 173 (71%) also had 1- and 12-month IOP data, respectively. The primary endpoint was absolute change in IOP (mmHg) at 1, 3, and 12 postoperative months relative to preoperative IOP at each timepoint. There were three secondary endpoints. (i) Identification of the perioperative factors that associated with IOP change at 3 months by univariable and multivariable analyses. These analyses were conducted with the 241 3-month whole-cohort patients because this timepoint associated with the strongest reduction in IOP. (ii) The 3-month whole cohort was categorized into three subgroups according to preoperative IOP thresholds of 20–29 (High, n = 49), 15–19 (Intermediate, n = 117), and 10–14 (Low, n = 75) mmHg. These thresholds were used previously in the literature [56]. The three preoperative-IOP subgroups were then compared in terms of change in IOP at 1, 3, and 12 months after cataract surgery. (iii) Multivariable analysis of each of the three preoperative-IOP subgroups was performed to determine the pre/perioperative factors that associated with IOP change at 3 months in each subgroup.

### Statistical analysis

Of the 292 phacoemulsifications in the RCT, 15 were converted to the non-randomized method for technical reasons. However, the intention-to-treat analysis of the original trial data also showed that subluxation was not inferior to DAC [62]. Thus, per-protocol trial data were analyzed in the current post-hoc analysis. The data were presented as mean±standard

deviation (SD) or n (%). There were no missing data for any variables. To test whether loss-to-follow-up at 12 months altered cohort demographic and clinical features, the 3-month whole-cohort (n = 241) and 12-month (n = 173) patients were compared in terms of pre/perioperative variables by Student's t test or Chi-squared test. To determine whether the IOP of the eyes changed significantly at 1, 3, and 12 months relative to baseline, Kruskall-Wallis analysis of variance (ANOVA) was conducted followed by Dwass, Steel, Critchlow-Fligner multiple comparison analysis. The latter is based on pairwise Wilcoxon comparisons. The three preoperative-IOP subgroups were compared in terms of IOP at each timepoint with Kruskall-Wallis ANOVA followed by Mann-Whitney U tests with Bonferroni correction. To determine the association between pre/perioperative variables and IOP change in the 3-month whole cohort, univariable analyses were conducted with Student's t-tests, ANOVA, or Pearson's correlation. For univariable analysis of implant power, implant power was categorized according to the 21 D threshold, which was the average power of the intraocular implants in the cohort. Multiple linear regression analyses were conducted with the whole 3-month cohort and each of the three preoperative-IOP subgroups with 10 independent variables, including ACD, AXL, and LT. To explore the role of LP and RLP, separate multivariable analyses were conducted with the same independent variables except for ACD and LT in the LP analysis (because LP = ACD + 0.5LT) and except for ACD, AXL, and LT in the RLP analysis (because RLP = LP/AXL). None of the independent variables in any multivariable analyses were collinear, as determined by Variance Inflation tests. To determine the effect of expressing IOP change as % change relative to baseline rather than as absolute IOP change in mmHg, preliminary multiple linear regression analysis with the 3-month cohort was repeated with percentage IOP change. To determine the effect of including 12-month data as well as 3-month data, we also conducted preliminary linear mixed-effects modeling with the 3- and 12-month cohorts that used time as an independent factor. All statistical analyses were performed with SAS software (version 9.3, SAS Inst., Cary, NC, USA). The significance threshold was set at 5%.

## Results

### Preoperative and operative characteristics of the cohort

Of the 292 patients included in the PERCEPOLIS trial, 22 (8%), 17 (6%), and 94 (32%) did not attend the 1-, 3-, and 12-month follow-up visit. Of the 270, 275, and 198 patients who attended these visits, 32, 34, and 25 were respectively excluded because IOP data were missing (n = 3, 2, and 0, respectively) and/or surgery had not been conducted by the principal investigator (n = 31, 32, and 25, respectively). Preoperative and 3-month postoperative IOP data were thus available for 241 patients (83% of the original cohort). Of these 'whole cohort' patients, 238 patients (99% of the whole cohort) also had 1-month postoperative IOP data while 173 (72%) also had 12-month postoperative IOP data (Fig 1). All patients were nonglaucomatous and did not receive any IOP-lowering treatment before or after surgery.

The average age±standard deviation of the whole cohort was 74±9 years and 56% were female. Half had intermediate (N3) cataract, a third had hard (N4/5) cataract, and the remaining fifth had soft (N1/2) cataract. Mean preoperative IOP was 17.6±3.5 mmHg. Mean preoperative ACD, AXL, LT, LP, and RLP were 3.1±0.4 mm, 23.2±0.8 mm, 4.6±0.4 mm, 5.4±0.4, and 0.23±0.01, respectively. Subluxation and DAC were performed in 52% and 48%, respectively. Mean EPT and implant power were 6±3 seconds and 21±4 D, respectively (Table 1). The 12-month patients (n = 173) did not differ significantly from the whole cohort (n = 241) in terms of any baseline/operative variables, including preoperative IOP (S4 Table).

### Change in IOP over time after surgery

At 1, 3, and 12 months, the average IOP of the eyes dropped from 17.6±3.5 mmHg preoperatively to 15.7±3.8 (−1.9±0.2), 15.3±3.0 (−2.4±0.2), and 15.9±3.2 (−1.7±0.2) mmHg, respectively (orange line in Fig 2). The drops at these timepoints relative to baseline were significant (all p < 0.001). The 12-month patients did not differ from the whole cohort in terms of absolute IOP drop at 3 months (−2.3±0.2 vs. −2.4±0.2 mmHg, p = 0.84 on Wilcoxon analysis).

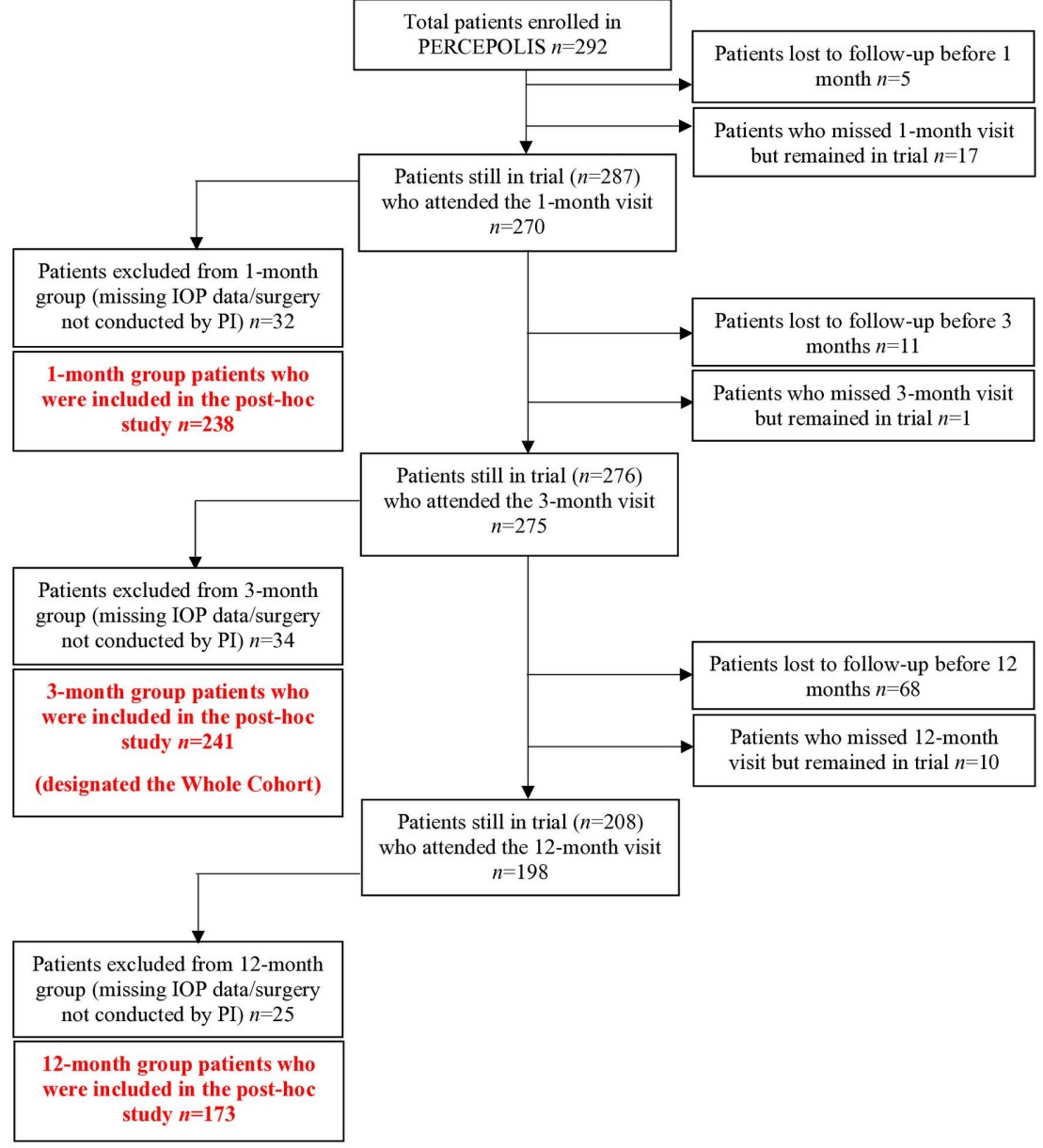

**Fig 1. Nonglaucomatous patient distribution in the PERCEPOLIS clinical trial (right boxes) and in the post-hoc analysis in the present study (left boxes).** IOP, intraocular pressure; PI, principal investigator.

Postoperative change in IOP can also be defined as % IOP change relative to baseline. The % change at 1, 3, and 12 months in our cohort was −11%, −14%, and −10%, respectively. Expressing IOP as % change did not markedly alter our study outcomes: for example, our preliminary multivariable analyses showed no marked differences when we used % change rather than absolute change (S5 Table). Moreover, when we converted the IOP data in the related literature to absolute or % IOP change, we observed identical trajectories of IOP over time for both IOP variables (S1 Fig). Since absolute IOP change is generally more clinically meaningful for clinicians than % IOP change, only absolute IOP change data are shown below.

**Table 1. Demographic, preoperative clinical, and operative characteristics of the 3-month whole cohort (n = 241).**

| Characteristic | Mean ± SD or n (%) |
|---|---|
| Age, years | 74 ± 9 |
| Female sex | 136 (56) |
| Cataract density | |
| N1/2 | 46 (19) |
| N3 | 116 (48) |
| N4/5 | 79 (33) |
| Preoperative IOP, mmHg | 17.6 ± 3.5 |
| Preoperative ACD, mm | 3.1 ± 0.4 |
| Preoperative AXL, mm | 23.2 ± 0.8 |
| Preoperative LT, mm | 4.6 ± 0.4 |
| Preoperative LP | 5.4 ± 0.4 |
| Preoperative RLP | 0.23 ± 0.01 |
| Surgical technique | |
| Subluxation | 125 (52) |
| DAC | 116 (48) |
| EPT, seconds | 6 ± 3 |
| Implant power, D | 21 ± 4 |

ACD, anterior chamber depth; AXL, axial length; DAC, divide-and-conquer; EPT, effective phaco time; IOP, intraocular pressure; LP, lens position (ACD + 0.5LT); LT, lens thickness; RLP, relative lens position (LP/AXL); SD, standard deviation.

The 3-month cohort not only had the most patients ($n = 241$), it also demonstrated the greatest reduction in IOP (−2.4 mmHg) (Fig 2). Thus, we focused on this cohort for our analyses searching for variables that associated with IOP change after cataract surgery. This approach was also supported by preliminary linear mixed-effects modeling that used time as an independent factor: when we compared this analysis to our standard linear-regression multivariable analysis, we found that time had a significant effect on postoperative IOP (beta = −0.72 ± 0.25, $r^2 = 0.01$, $p = 0.003$), and its inclusion in the multivariable analysis completely abolished the significance of one independent variable (age: beta = 0.04 ± 0.02, $r^2 = 0.01$, $p = 0.04$ in the standard analysis changed to beta = 0.0002 ± 0.02, $r^2 < 0.001$, $p = 0.99$ in the mixed-effects analysis) (S6 Table). This suggested that as time progresses, the mechanisms that shape postoperative IOP may change. Thus, to help interpret the data, we focused on one cross-sectional timepoint, namely, 3 months.

## Association between IOP change in the 3-month whole cohort and 12 pre/perioperative variables

On univariable analysis, preoperative IOP correlated strongly and positively with 3-month IOP change (r = 0.60, 95% confidence intervals −0.67 to −0.51, $p < 0.0001$) (S7 Table and S2 Fig). Age, sex, cataract density, preoperative ACD, AXL, LT, LP, and RLP, surgical technique, EPT, and implant power did not associate with IOP change (S7 Table).

On multiple linear regression with 10 independent variables, including preoperative IOP, 3-month IOP change was much greater when preoperative IOP was high (beta = 0.53 ± 0.04, $p < 0.001$). Preoperative IOP accounted for 36% of the IOP variance. Male sex also predicted greater IOP drop (beta = 0.79 ± 0.31, $r^2 = 0.02$, $p = 0.01$). Older age also tended to predict greater IOP drop (beta = 0.04 ± 0.02, $r^2 = 0.01$, $p = 0.052$). Cataract density, preoperative ACD, AXL, and LT, type of surgery, EPT, and implant power did not predict IOP drop in the whole cohort (Table 2). Separate multivariable analyses with LP and RLP showed they also did not predict IOP change (S8 Table).

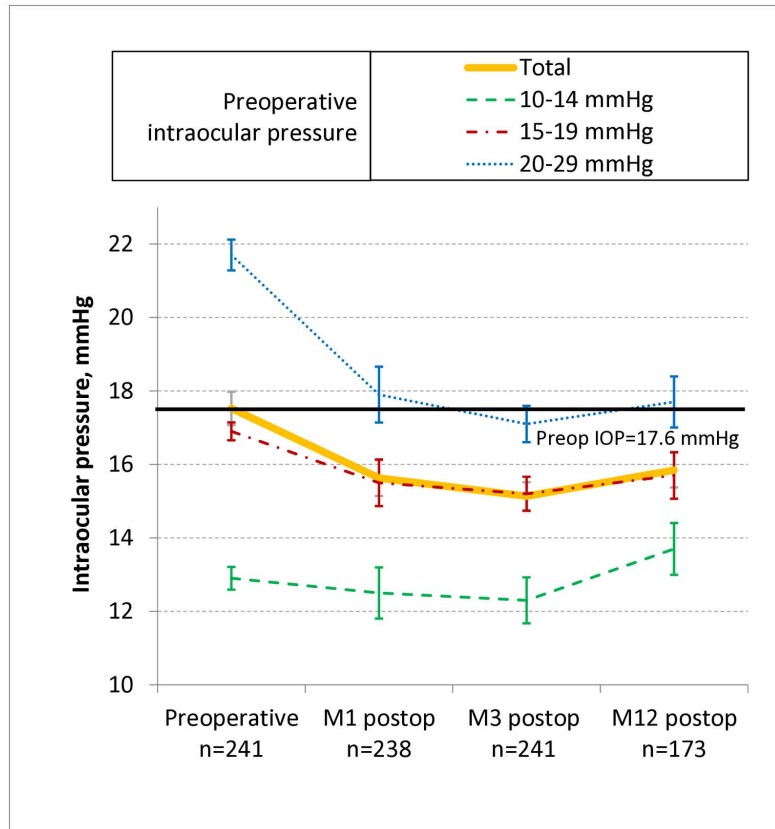

**Fig 2. Change in IOP at 1, 3, and 12 months in the whole cohort and the three preoperative-IOP subgroups.** The average IOP of the patients dropped significantly at the 1-, 3-, and 12-month timepoints relative to baseline (orange line), as indicated by Kruskall-Wallis ANOVA followed by Dwass, Steel, Critchlow-Fligner multiple comparison analysis (all p<0.001). The three preoperative-IOP subgroups in the 3-month whole cohort (blue, red, and green lines) differed significantly from each other in terms of their preoperative, 1-month, 3-month, and 12-month IOPs, as indicated by Kruskall-Wallis ANOVA followed by Mann-Whitney U tests with Bonferroni correction (all p<0.001). M, month; IOP, intraocular pressure; postop, postoperative.

### IOP change in subgroups with High, Intermediate, and Low preoperative IOP

Our whole-cohort multivariable analysis showed that high preoperative IOP strongly predicted a greater IOP fall at 3 months (Table 2). We examined this relationship between preoperative and postoperative IOP more closely by dividing the 3-month whole cohort into three subgroups according to their preoperative IOP, namely, the High (20–29 mmHg, $n=75$), Intermediate (15–19 mmHg; $n=117$), and Low (10–14 mmHg; $n=49$) subgroups. Comparison of the three subgroups in terms of IOP change over time confirmed that the High subgroup showed the greatest IOP reduction over time, followed by the Intermediate subgroup and then the Low subgroup. In all subgroups, the drop at 1 and 3 months was followed by a slight increase in IOP at 12 months (Fig 2). The differences between the subgroups in IOP were statistically significant at all time points (all $p<0.001$).

### Pre/perioperative predictors of IOP change in the three preoperative-IOP subgroups

Cataract surgery is thought to lower IOP in PACG and POAG by widening the iridotrabecular angle and possibly by remodeling the trabecular meshwork, respectively. However, the mechanisms by which cataract surgery reduces IOP in nonglaucomatous patients remain poorly understood [3,36]. Given the strong relationship between preoperative and

**Table 2. Multiple linear regression analysis of the ability of pre/perioperative variables to predict absolute IOP change (mmHg) at 3 months ($n = 241$, $r^2 = 0.40$).**

| Variable | Beta ± SD | Partial r² | p |
|---|---|---|---|
| Age, years | 0.04 ± 0.02 | 0.01 | 0.052 |
| Male sex | 0.79 ± 0.31 | 0.02 | **0.01** |
| Cataract density | | 0.002 | |
| N1/2<br>N3<br>N4/5 | Ref.<br>−0.25 ± 0.43<br>−0.38 ± 0.46 | | Ref.<br>0.56<br>0.41 |
| Preoperative IOP, mmHg | 0.53 ± 0.04 | 0.36 | **<0.001** |
| Preoperative ACD, mm | 0.68 ± 0.47 | 0.005 | 0.15 |
| Preoperative AXL, mm | 0.03 ± 0.24 | <0.001 | 0.88 |
| Preoperative LT | 0.11 ± 0.46 | <0.001 | 0.81 |
| Subluxation surgery | 0.21 ± 0.31 | 0.001 | 0.50 |
| EPT, seconds | −0.04 ± 0.05 | 0.001 | 0.40 |
| Implant power, D | 0.01 ± 0.04 | <0.001 | 0.67 |

ACD, anterior chamber depth; AXL, axial length; EPT, effective phaco time; IOP, intraocular pressure; LT, lens thickness; ref., reference; SD, standard deviation.

postoperative IOP in our patients (Table 2), we explored the underlying mechanisms in nonglaucomatous patients by subjecting the High, Intermediate, and Low subgroups of the 3-month whole cohort to the same multivariable analyses described above. We chose to use separate multivariable analyses per subgroup rather than a single multivariable analysis with preoperative IOP divided into High/Intermediate/Low categories because our approach avoided the pitfall of a possible interaction between preoperative and postoperative IOP.

The multivariable analyses showed that higher preoperative IOP predicted a larger absolute IOP change at 3 months in particularly the High subgroup (beta = 0.72 ± 0.15, $r^2 = 0.19$, p < 0.0001) but also the Intermediate (beta = 0.36 ± 0.18, $r^2 = 0.03$, p = 0.045) and Low (beta = 0.61 ± 0.30, $r^2 = 0.07$, p = 0.0505) subgroups (Table 3). Notably, the three subgroups differed in terms of the other variables that predicted IOP change. In the High subgroup, male sex predicted greater IOP change (beta = 1.63 ± 0.57, $r^2 = 0.07$, p = 0.005). There was also a tendency for older age and lower implant power to predict IOP change in this subgroup (beta = 0.06 ± 0.04, $r^2 = 0.03$, p = 0.08 and beta = 0.11 ± 0.06, $r^2 = 0.03$, p = 0.09, respectively). In the Intermediate subgroup, higher implant power predicted greater IOP change (beta = 0.19 ± 0.07, $r^2 = 0.06$, p = 0.01). In the Low subgroup, harder cataracts associated with less IOP change relative to moderate cataract density (beta = 2.14 ± 0.89, $r^2 = 0.13$, p = 0.02). Type of surgery, preoperative ACD, AXL, and LT, and EPT were not predictive in any subgroup (Table 3). Separate multivariable analyses with RLP instead of ACD, AXL, and LT did not find that RLP predicted preoperative IOP change in any subgroup (S9 Table).

## Discussion

### Cataract surgery significantly reduces IOP in nonglaucomatous patients

This post-hoc analysis of PERCEPOLIS trial data showed that in eyes without glaucoma and no IOP-lowering treatment before or after surgery, phacoemulsification significantly reduced IOP: at 3 months, the average IOP fell by 2.4 ± 0.2 mmHg (14%). This is consistent with previous studies that examined the change in IOP in nonglaucomatous eyes after phacoemulsification (S1 Table, S1 Fig) [13–35].

**Table 3.** Multiple linear regression analysis of the ability of pre/perioperative variables to predict absolute IOP drop at 3 months in eye subgroups with preoperative IOP of 20–29, 15–19, or 10–14 mmHg (total *n*=241).

| Variable | Preoperative IOP 20–29 mmHg (High subgroup) *n*=75[a] | | | Preoperative IOP 15–19 mmHg (Intermediate subgroup) *n*=117[a] | | | Preoperative IOP 10–14 mmHg (Low subgroup) *n*=49[a] | | |
|---|---|---|---|---|---|---|---|---|---|
| | Beta±SD | Partial r² | p | Beta±SD | Partial r² | p | Beta±SD | Partial r² | p |
| Age, years | 0.06±0.04 | 0.03 | 0.08 | 0.01±0.03 | 0.001 | 0.78 | 0.02±0.05 | 0.004 | 0.61 |
| Male sex | 1.63±0.57 | 0.07 | **0.005** | 0.53±0.47 | 0.01 | 0.26 | 0.32±0.69 | 0.004 | 0.64 |
| Cataract density | | 0.01 | | | 0.02 | | | 0.13 | |
| N1/2<br>N3<br>N4/5 | 0.83±0.70<br>Ref.<br>−0.59±0.62 | | 0.25<br>Ref.<br>0.34 | 0.79±0.73<br>Ref.<br>−0.07±0.51 | | 0.28<br>Ref.<br>0.90 | −1.15±0.78<br>Ref.<br>−2.14±0.89 | | 0.15<br>Ref.<br>**0.02** |
| Preop IOP, mmHg | 0.72±0.15 | 0.19 | **<0.0001** | 0.36±0.18 | 0.03 | **0.045** | 0.61±0.30 | 0.07 | **0.0505** |
| Preop ACD, mm | 0.33±0.78 | 0.002 | 0.67 | 0.60±0.73 | 0.01 | 0.42 | 1.27±1.06 | 0.03 | 0.24 |
| Preop AXL, mm | 0.12±0.40 | 0.001 | 0.76 | −0.04±0.34 | <0.001 | 0.91 | −0.43±0.60 | 0.01 | 0.48 |
| Preop LT, mm | −0.80±0.86 | 0.01 | 0.35 | 0.54±0.73 | 0.01 | 0.46 | −0.20±0.85 | 0.001 | 0.82 |
| Subluxation surg | 0.19±0.54 | 0.001 | 0.72 | 0.02±0.46 | <0.001 | 0.97 | 1.02±0.69 | 0.04 | 0.15 |
| EPT, seconds | −0.11±0.09 | 0.01 | 0.25 | −0.06±0.08 | 0.01 | 0.44 | 0.14±0.12 | 0.02 | 0.27 |
| Implant power, D | −0.11±0.06 | 0.03 | 0.09 | 0.19±0.07 | 0.06 | **0.01** | −0.04±0.10 | 0.003 | 0.70 |

ACD, anterior chamber depth; AXL, axial length; EPT, effective phaco time; IOP, intraocular pressure; LT, lens thickness; preop, preoperative; Ref., reference; SD, standard deviation; surg, surgery.

[a]r²=0.46, 0.13, and 0.33 for the model in the High, Intermediate, and Low subgroups, respectively.

## Change in IOP over time

The orange line in Fig 2 shows that on average, IOP in our patients dropped, stabilized, and then rose slightly. This pattern was also observed in previous studies on nonglaucomatous eyes (S1 Fig). Thus, the beneficial effect of cataract surgery seems to wane over time. However, studies with follow up at 2–3 years suggest that IOP generally does not return to preoperative values (S1 Fig). The two longest-term studies provide conflicting results: Suzuki et al. observed that the IOP change in nonglaucomatous patients was +0.1 mmHg (+1%) at 6 years [56] whereas Shingleton et al. noted an ongoing IOP reduction at 5 years (−1.5 mmHg, −9%) [22]. It should be noted that studies on the long-term outcomes of an intervention inevitably suffer from loss to follow-up, and this is particularly true for studies in older populations such as cataract-surgery patients. Indeed, 28% of our whole cohort was lost to follow up at the 12-month visit. Similar loss-to-follow-up rates were also observed by other studies on IOP change after phacoemulsification in nonglaucomatous eyes (34% and 37% at 1 year) [20,21]. This together with the often low sample sizes at later timepoints and the paucity of studies on long-term IOP outcomes in nonglaucomatous patients means it remains unclear how long the beneficial effects of cataract surgery on IOP are sustained in these patients.

**Glaucoma patients.** Glaucoma patients also display a marked drop in IOP after cataract surgery. When we briefly assessed the previous studies that directly compared nonglaucomatous patients to patients with PACG, POAG, pseudoexfoliation-associated glaucoma (PXFG), or unspecified glaucoma [14,21,22,27–29,34,41,78,79], we found that the glaucoma eyes generally had higher average preoperative IOP than the nonglaucomatous eyes (range: 14.3–22.1 vs. 13.6–16.5 mmHg), even though most were being treated with anti-glaucoma medication. Nonetheless, their postoperative IOP changes followed the same pattern, namely, an initial sharp rise, followed by a marked and extensive drop, and then a gradual rise (S3 Fig). When directly compared to nonglaucomatous eyes, PACG [78,79] and PXFG [21,29] eyes had significantly greater IOP drops. However, this was not observed for the POAG and unspecified glaucoma groups [14,22,27,28,34,41,79]. Notably, the drop in IOP in our cohort was similar to that observed in the nonglaucomatous and POAG groups (S3 Fig). These findings are consistent with the general literature on glaucoma: a recent meta-analysis of

34 studies reported that PACG, PXFG, and POAG associate with a respective post-cataract surgery decrease in IOP of −7.0, −5.3, and −2.7 mmHg at 12 months [12]. A 2018 meta-analysis on 32 studies also reported similar IOP drops for PACG, PXG, and POAG at ≥12 months (−6.4, −5.8, and −2.7 mmHg, respectively) [9]. The greater IOP reduction in PACG compared to POAG was also observed in a Baynesian analysis of 13 studies [80]. Thus, PACG and PXFG eyes exhibit greater IOP falls than POAG or nonglaucomatous eyes, which have similar IOP drops. Notably, cataract surgery in the glaucoma eyes also associated with a significant reduction in anti-glaucoma medication use in four [14,27,31,34] of the five [22,79,81] studies that examined this variable: this was observed for both PACG and POAG [14,31,34] and indicates that the post-cataract surgery IOP drop in glaucoma is clinically significant.

**Higher preoperative IOP predicts greater falls in IOP**

Our study and previous studies thus show that in both nonglaucomatous and glaucomatous eyes, IOP drops after cataract surgery for at least a few years. Fig 2 and S2 Fig demonstrated that the drop in IOP by nonglaucomatous patients was greatest in patients with a high baseline IOP (20–29 mmHg). This has also been observed repeatedly in studies on nonglaucomatous cohorts (Fig 3) [33,47,50,54,56,78]. In addition, multivariable analysis with our 3-month whole cohort showed that higher preoperative IOP predicted a larger IOP drop after cataract surgery (beta = 0.53 ± 0.04, p < 0.001). All previous studies that included preoperative IOP as an independent variable in univariable/multivariable analyses also found that it correlated strongly with or predicted IOP drop (S2 Table) [15,17,18,29,32–34,39,41,43–49,53–58,82–84].

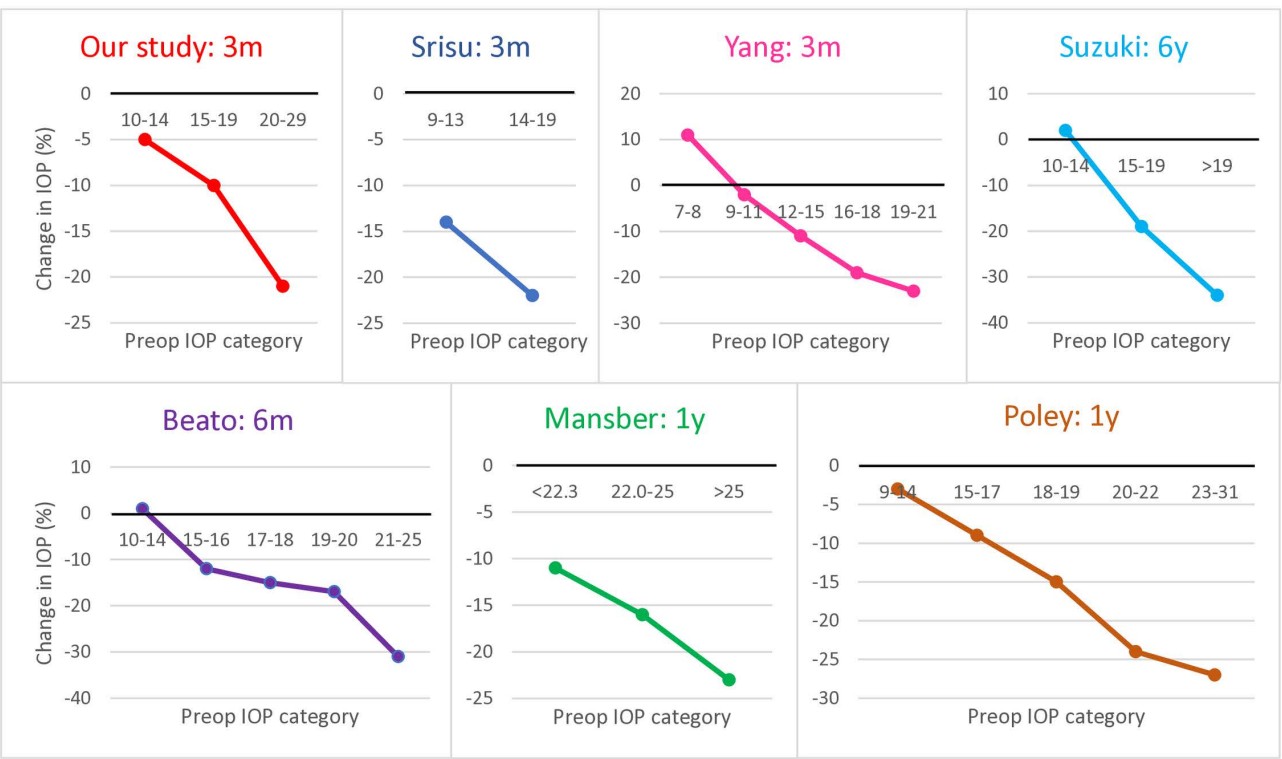

**Fig 3. IOP change according to preoperative IOP category in our study and previous studies.** The studies were by Srisuwanporn et al. [78], Yang et al. [47], Suzuki et al., Beato et al. [33], Mansberger et al. [50], and Poley et al. [54]. In the study cohort of Mansberger et al., all eyes were nonglaucomatous but had ocular hypertension. The remaining cohorts consisted of normotensive nonglaucomatous eyes only. IOP, intraocular pressure; m, month; y, year.

Specifically, these studies showed that on univariable analysis, the regression coefficient between preoperative and postoperative IOP ranges from 0.25 to 0.75 (our r = 0.60), and on multivariable analysis, postoperative IOP decreases by 0.22 to 0.60 mmHg for every 1 mmHg-higher preoperative IOP (our beta = 0.53). The positive correlation between higher preoperative IOP and IOP drop after cataract surgery has also been observed for glaucoma patients [11,85]. These findings indicate that preoperative IOP is a very strong positive predictor of IOP after cataract surgery: indeed, it accounted for 36% of the total postoperative IOP variance in our 3-month whole cohort (Table 2).

### Regression-to-the-mean

It has been speculated that the IOP drop after cataract surgery reflects the misleading statistical phenomenon called regression-to-the-mean (RTM) [9–11]. IOP shows considerable variability due to instantaneous fluctuations (caused for example by saccades), short-term fluctuations over days/weeks (caused for example by diurnal variation of 2–3 mmHg [86]), and long-term fluctuations over months to years (caused for example by the onset or progression of disease) [87–89]. Thus, a single IOP measurement of a patient that is taken at baseline may be quite disparate from the natural average IOP of the patient. If the baseline measurement is abnormally high or low relative to the natural mean, subsequent measurements will tend to regress towards the mean, meaning that patients with an unnaturally high baseline IOP value will tend to show a drop in IOP over time while patients with an unnaturally low baseline IOP value will show a rise. RTM is likely to be particularly significant when glaucoma patients are selected for inclusion into a study on the basis of a single baseline IOP measurement, but it will also operate in unselected nonglaucomatous cohorts such as those in ours. Indeed, Fig 2 suggested that RTM was a factor in the IOP drop of our patients, since IOP rose from 15.3 mmHg at 3 months to 15.9 mmHg at 12 months.

Since RTM can mimic treatment effects, RTM must be excluded by careful study design [90]. In particular, baseline IOP should be measured at the same time of day according to standard protocols and most importantly, measured several times over the months before study commencement. The latter approach has very rarely been adopted in the literature on the effect of cataract surgery on IOP. We also did not adopt it because our primary RCT objective was endothelial-cell loss.

To ascertain whether the IOP drop after cataract surgery that we observed in our nonglaucomatous patients is a real biological phenomenon, we collected the few studies that made an effort to counteract RTM or identify how much it contributed to the perceived fall in IOP. These are summarized in S10 Table and depicted in S4 Fig. A particularly important study was the Ocular Hypertension Study on hypertensive eyes that did not exhibit signs of glaucoma. This study employed two measures to account for RTM: (i) it compared 63 eyes that underwent cataract surgery to 743 baseline IOP-matched eyes that did not undergo cataract surgery, and (ii) baseline IOP was the average of 3 measurements made at 6-monthly intervals over the 1.5 years before cataract surgery or inclusion in the study. The intervention eyes demonstrated a significant and sustained drop in IOP over time: at 6 months, IOP was −3.9 mmHg (−17%). By contrast, the control eyes evinced no change [50]. Qassim et al. reported a similar prospective study on 171 eyes with probable early-stage glaucoma. This study included 171 control eyes that were matched for age, sex, and baseline IOP, and preoperative IOP of all eyes was based on 7–11 preoperative readings in the preceding 2 years. Cataract surgery reduced IOP by −2.2 mgHg at 6 months whereas the control eyes demonstrated an increase of +0.6 mmHg at the same timepoint [6]. Three other studies also used multiple preoperative IOP measurements. Thus, the prospective study of Ramli et al. on 86 nonglaucomatous eyes measured baseline IOP on three different occasions and found the IOP dropped by −1.8 mmHg at 1 month [32]. Similarly, Markic et al. measured baseline and postoperative IOP at 07:30, 13:30, and 19:30 in 1 day and found that the IOP of 31 nonglaucomatous eyes dropped by −2.4 mmHg at 3 months [29]. Moreover, Pradhan et al. showed that when four preoperative IOP values from the previous 5 years were averaged, cataract surgery in 77 patients (77% nonglaucomatous, 13% with POAG/suspected POAG) reduced IOP by −3.2 mmHg at 2–3 months. This study also showed that a single preoperative IOP value (the most recent) accounted for only 13% of the variation in postoperative

IOP but averaging the four preoperative IOPs increased this to 20% [49]. Similar results were reported when the unoperated fellow eye served as a control. This is a reasonable approach since right and left eye IOPs correlate quite strongly (r = 0.7) [18,87]. Thus, four studies on normotensive nonglaucomatous eyes that underwent cataract surgery (*n* = 29, 117, 999, and 42, respectively) showed that IOP in the operated eye dropped by −1.2, −2.5, −2.4, and −1.6 mmHg at 1 [83], 1 [46], 1 [47], and 3 months [91], respectively. By contrast, the contralateral eye demonstrated IOP changes of +0.3, −0.2, −0.2, and −0.4 mmHg at these timepoints, respectively [46,47,83,91]. Similarly, in PACG eyes, cataract surgery reduced IOP by −3.5 mmHg at 3 months compared to −0.2 mmHg in the fellow eyes [92].

In summary, the six studies that controlled for RTM and examined normotensive nonglaucomatous eyes suggest that cataract surgery reduces IOP in these eyes by −1.2 to −2.5 mmHg at 1–3 months, similar to the IOP changes of −2.0 and −2.4 mmHg at 1 and 3 months that we observed, respectively. By contrast, contralateral-eye IOP changes by +0.3 to −0.4 mmHg at these timepoints [29,32,46,47,83,91]. Thus, RTM may account for only a small proportion of the IOP drop after cataract surgery, suggesting that cataract surgery induces a biologically real and clinically significant change in IOP.

## Associations between postoperative IOP drop and pre/perioperative variables differ depending on preoperative IOP

In glaucoma, the mechanisms by which cataract surgery improves IOP may differ depending on the type of disease: in angle closure, which associates with particularly high IOP, the IOP drop seems to due to angle opening, whereas in open angle glaucoma, the IOP drop may reflect remodeling of the trabecular meshwork [3]. However, the mechanisms that drive IOP drop in nonglaucomatous eyes after cataract surgery are not known. To elucidate these mechanisms, we divided the whole cohort into three subgroups on the basis of their preoperative IOP (High, Intermediate, and Low) and searched for potential predictors and mechanisms shaping their change in IOP. Multivariable analyses showed that while preoperative IOP continued to predict postoperative IOP change in the High (beta = 0.72 ± 0.15, $r^2$ = 0.19, p < 0.0001), Intermediate (beta = 0.36 ± 0.18, $r^2$ = 0.06, p = 0.045), and even Low (beta = 0.61 ± 0.30, $r^2$ = 0.07, p = 0.0505) subgroups, the subgroups differed in terms of other variables that predicted postoperative IOP drop. Specifically, higher implant power, male sex, and older age associated with IOP drop in the High subgroup; lower implant power associated with IOP drop in the Intermediate subgroup; and greater cataract density associated with IOP drop in the Low subgroup. This suggests that different mechanisms may underpin the IOP drop in these subgroups. These potential mechanisms are explored in following subsections.

**Implant power.** Higher power implants predicted greater IOP drop in the Intermediate preoperative-IOP subgroup (beta = 0.19 ± 0.07, $r^2$ = 0.06, p = 0.01) while the High IOP subgroup tended to show the opposite pattern, namely, lower power implants associated with greater IOP drop (beta = 0.11 ± 0.06, $r^2$ = 0.03, p = 0.09). Implant power is a broad measure of ocular macroanatomy: eyes that receive high power implants are smaller and have more crowded anterior chambers whereas eyes treated with low power implants are longer and have more capacious anterior chambers. Notably, hyperopia associates with angle closure and PACG [93] whereas myopia associates with all forms of open-angle glaucoma [93,94]. Thus, the overall ocular anatomy of the eyes with Intermediate- and High-preoperative IOP may tend to resemble that of patients with PACG and POAG, respectively. This suggests that the IOP drop in the Intermediate- and High-IOP subgroups may be driven by the same distinct mechanisms that are thought to lead to these pathologies, namely, angle closure and trabecular-meshwork resistance, respectively.

Notably, implant power did not predict IOP drop in the whole cohort, similar to a similar study on nonglaucomatous eyes [15] (S2 Table). This may reflect the opposite effects of implant power in the Intermediate and High subgroups.

**ACD, AXL, LT, LP, and RLP.** We did not find that preoperative ACD, AXL, LT, or composites of these ocular macroanatomy variables (LP and RLP) predicted IOP drop in any of the preoperative IOP subgroups or the whole cohort. Many studies have assessed the relationship between IOP drop after cataract surgery and these and other macroanatomical variables (e.g., anterior-chamber angle, angle-opening distance, iris curvature and thickness, and lens vault) in the hope of finding variables that can predict IOP outcomes (S3 Table) [15,17,29,30,32–34,41,44–48,52,57,58,95].

This research reflects the fact that cataract surgery reverses the macroanatomical changes induced by cataract, namely, the abnormal protrusion of the lens into the anterior chamber, anterior bowing of the iris, anterior shifting of the scleral spur, ACD shallowing, AXL shortening, narrowing of the iridotrabecular angle, and compression and anterior displacement of the ciliary body [96–98]. The ability of cataract surgery to restore the posterior localization of the ciliary body may be particularly important because several studies show that this likely renews the traction by the ciliary body on the trabecular meshwork and Schlemm canal, which in turn relieves iridotrabecular angle compression [98,99] and increases outflow [91,100]. However, while some studies show associations between specific ocular macroanatomical variables and post-surgical IOP, many others, like us, do not. This inter-study variability is observed in both glaucomatous (including PACG) and nonglaucomatous eyes (S3 Table). It is possible that these macroanatomical variables do not adequately capture all the ocular changes that promote ocular hypertension in cataracted eyes and that conversely lead to IOP drop after cataract surgery. Indeed, it is thought that cataract surgery also induces important microanatomical changes to the ciliary body, trabecular meshwork, Schlemm canal, and uveoscleral-pathway, possibly by inducing the ocular tissue to release prostaglandins. These mediators may provoke metalloproteinase-mediated remodeling of the ocular tissues, thereby improving outflow-tissue function and possibly also reducing aqueous-humor production by the ciliary body [34,101–105].

There may also be another reason why macroanatomical measurements did not predict IOP change after cataract surgery in our study: we included implant power in our multivariable model. Implant power is not only a broad macroanatomical variable that encompasses ACD and AXL, it is also a postoperative variable, unlike ACD and AXL in our study (and most other studies (S3 Table)). Thus, implant power may represent the macroanatomical changes in the eye after cataract surgery more fully than specific preoperative variables such as ACD and AXL. Consequently, the multivariable model in our study may have allocated any ACD/AXL-related IOP variance to implant power.

**Sex.** We found that men showed greater IOP drop than women in the High-preoperative IOP subgroup (beta = 1.63 ± 0.57, $r^2$ = 0.07, p = 0.005) but not the other subgroups. Sex could potentially shape IOP drop after cataract surgery via several mechanisms. The first relates to hormones: at least in women, serum testosterone and estradiol correlate positively with IOP [106,107], and post-menopausal hormone-replacement therapy and pregnancy reduce IOP [108]. Moreover, while the role of testosterone in IOP is still poorly understood [108], the ciliary epithelium, trabecular meshwork, and blood vessels bear estrogen receptors, the activation of which may reduce aqueous-humor production, improve outflow pathways, and decrease episcleral venous pressure [109]. The second possible reason is anatomical: men have deeper anterior chambers and wider anterior-chamber angles and demonstrate less anterior chamber shallowing with age [110,111]. Thus, sex may play a small role in shaping IOP drop after cataract surgery in nonglaucomatous eyes with high preoperative IOP.

The association between greater IOP drop in men after cataract surgery was also observed in our whole cohort (beta = 0.79 ± 0.31, $r^2$ = 0.02, p = 0.01). In the literature, greater IOP drop associated with male sex on four univariable (analyses [34,41,45,52] and female sex on univariable analysis in a registry study on 20437 eyes [44], while 11 other studies showed no effect of sex (S2 Table). This interstudy variability may partly reflect our finding that sex only associates with IOP drop in a subsection of patients.

**Cataract density.** Greater cataract density predicted a larger IOP drop in the Low-preoperative IOP subgroup (beta = 2.14 ± 0.89, $r^2$ = 0.13, p = 0.02) but not the other subgroups. This did not reflect different cataract density distributions between the three subgroups (p = 0.12, Chi-squared test). The reason for this difference between the subgroups is unclear. One possibility is that hard cataract acted as a surrogate for an operative variable such as high infusion volume: it has been suggested that high infusion volumes help clear out clogged trabecular meshworks, thus promoting conventional outflow [21,39,43,73]. However, it is unclear why such a mechanism would operate in only the Low-preoperative IOP subgroup.

Cataract density was not a predictor of IOP drop in our whole cohort, and five other studies have also not noted an association [15,33,34,42,55] (S2 Table). Again, this may reflect our finding that this association was only observed in a subset of patients.

**Age.** We found that older age tended to predict IOP drop in the High preoperative-IOP subgroup (beta = 0.06 ± 0.04, $r^2$ = 0.03, p = 0.08) but not the other subgroups. How age could influence IOP is unclear since the literature suggests that older age can both decrease aqueous humor production [112] while conversely increasing outflow pathway resistance [113]. Which of these mechanisms prevails may reflect IOP-shaping ethnosocial differences such as the prevalence of myopia [114], obesity, and/or genetics [115].

We also observed that older age tended to predict IOP drop in our whole cohort (beta = 0.04 ± 0.02, $r^2$ = 0.01, p = 0.052). This association was also reported by the large registry study [44], whereas another study found that younger age associated with greater IOP drop [34] and 19 other studies found no relationship (S2 Table). Notably, we observed that age disappeared as a predictive variable when time in the form of the 12-month data was included in multivariable analysis (S6 Table), which suggests that this factor has only a transient effect on IOP after cataract surgery. Thus, age may play only a small role in the change in IOP after cataract surgery, and only in eyes with high preoperative IOP.

### Pre/perioperative variables that did not predict postoperative IOP drop

**EPT.** It has been suggested that the heat and/or shockwaves from phacoemulsification promote stress responses by trabecular-meshwork cells, which then induce tissue remodelling and favorable ocular microanatomical and/or biological changes [116]. This possibility is supported by two studies, which found that longer phaco time (a component of EPT) associates with more IOP drop [15,24]. However, we and five other studies did not observe relationships between EPT and IOP drop (S2 Table). Moreover, FLACS, which associates with significantly less EPT than conventional phacoemulsification, does not associate with less IOP drop than phacoemulsification [13,117]. This is also true for manual small-incision cataract surgery, which does not involve ultrasound [17,18]. Thus, at best, this ultrasound-related mechanism likely plays a very minor role in the IOP drop after phacoemulsification.

**Type of surgery.** We did not find that the method of cataract fragmentation affected 3-month IOP drop in either the whole cohort or the subgroups. This has also been observed when comparing conventional phacoemulsification to other methods of cataract extraction, including FLACS [13,117] and manual small-incision cataract surgery [17].

### Study limitations

The advantages of this study include the fact that none of the patients received IOP-lowering medications before or after surgery. It also benefitted from its randomized controlled design and the strict and rigorously enforced protocol for determining IOP. In addition, a wide range of cataract densities was included. However, there were some study limitations. First, we lacked a control for RTM, because the outcome variable of the original RCT was endothelial-cell loss, not IOP. Nonetheless, our examination of the literature suggested that while RTM does contribute to the perceived IOP drop after cataract surgery, this contribution is relatively small (S4 Fig and S10 Table). Second, we did not examine other ocular macroanatomical variables because they were not routinely recorded in the RCT. Of particular interest would be lens vault, iris area, and angle variables such as angle-opening distance: all have been shown to associate with IOP drop in nonglaucomatous and/or glaucomatous patients in some studies (S3 Table). Third, it is possible that more precise measurements of ocular macroanatomical and especially microanatomical variables, such as phase-sensitive OCT measurements of the trabecular meshwork, could find associations between these variables and IOP change. Fourth, 28% of the whole cohort dropped out at 12 months. Thus, the 12-month IOP data in Fig 2 should be interpreted cautiously. However, the 12-month patients did not differ from the whole cohort in terms of baseline variables or IOP drop at 3 months (S4 Table). Fifth, for all analyses, we defined change in IOP as absolute change in mmHg relative to baseline, not as % IOP change. However, our preliminary multivariable analyses showed no marked differences when we used % change rather than absolute change (S5 Table). Moreover, most other studies on nonglaucomatous eyes also used this definition (S2 Table) and the few studies that compared % IOP change to absolute IOP change did not find substantial differences in the results [41,45]. In addition, when we converted the IOP data in the related literature to absolute or % IOP change,

we observed identical trajectories of IOP over time for both IOP variable types (S1 Fig). Sixth, the relatively short follow up meant that we could not determine how long the IOP drop in our patients was sustained. Seventh, we focused on the 3-month data only for the multivariable analyses. However, this was supported by the fact that the IOP drop was most pronounced at that stage. Moreover, when we compared linear mixed-effects modeling that used time as an independent factor to our standard linear-regression multivariable analysis, we found that time had a significant effect on postoperative IOP and its inclusion in the multivariable analysis completely abolished the significance of age (S6 Table). This suggested that as time progresses, the mechanisms that shape postoperative IOP may change. This seems plausible. For example, while wound-healing remodeling may permit the ocular fluid to flow well at 3 months, this remodeling may have stopped at 12 months and mechanisms that impede flow have either returned, now predominate, or have newly emerged. To help interpret the data, we therefore focused on one cross-sectional timepoint, namely, 3 months. Eighth, one experienced surgeon conducted all surgeries included in the post-hoc analysis; while this limited surgeon-related confounding, it means that our findings cannot necessarily be extrapolated to other settings. Ninth, the patients were French, meaning that our findings reflect the sociodemographics of the French population. The findings may vary with other populations, particularly Asian patients, who are more prone to angle closure [118]. Finally, the sample size was limited, meaning that we would only be able to identify large associations with a high degree of certainty.

## Conclusions

We showed that 3 months after cataract surgery, IOP in nonglaucomatous eyes dropped by a clinically relevant 2.4 mmHg, consistent with previous studies on nonglaucomatous eyes (S1 Table, S1 Fig). The IOP-lowering effect of cataract surgery was most pronounced for the eyes with high preoperative IOP. While this is suggestive of the clinically misleading statistical phenomenon called RTM, our examination of studies that controlled for RTM suggested that only a small portion of the IOP-lowering effect of cataract surgery in nonglaucomatous eyes is due to RTM. Thus, the IOP-lowering effect of cataract surgery in nonglaucomatous eyes may be biologically real and clinically relevant.

Our data confirm that the most reliable predictor of IOP drop after cataract surgery is the preoperative IOP: clinicians can expect that at 3 months, nonglaucomatous eyes with preoperative IOPs of 20–29, 15–19, and 10–14 mmHg will experience an IOP drop of 4.6 (95%CI = 0.5–5.1), 1.7 (0.5–2.2), and 0.6 (0–1.2) mmHg, respectively (Fig 2).

Our multivariable analyses on the High, Intermediate, and Low preoperative-IOP subgroups suggested that phacoemulsification surgery lowers IOP in these subgroups via distinct mechanisms. Specifically, the surgery may reduce IOP in eyes with intermediate preoperative IOP by widening the iridocorneal angle (similar to PACG eyes) whereas the IOP reduction in the eyes with high preoperative IOP may be due to more subtle changes in the systems that produce and drain the aqueous fluid (similar to POAG eyes). These findings together suggest that different IOP-regulating mechanisms operate in even normotensive eyes. Further studies are needed to determine (i) the macroanatomical and microanatomical mechanisms by which cataract surgery lowers IOP in nonglaucomatous and glaucomatous eyes, (ii) how these mechanisms interact to shape postoperative IOP, and (iii) whether their influence changes over time after surgery.

The fact that cataract surgery lowers IOP in nonglaucomatous eyes in a clinically meaningful and apparently relatively persistent manner does not mean that cataract surgery should be performed for this reason alone. Nonetheless, our study shows that this IOP-lowering effect may be a real biological phenomenon that should be part of clinical decision-making, particularly when dealing with hypertensive nonglaucomatous eyes.

## Supporting information

**S1 Fig. Studies showing Absolute (mmHg) (A) and % (B) change in IOP over time in nonglaucomatous patients after cataract surgery.** All studies used phacoemulsification to remove the lens [13–21,23–25,27,29–35]. Our data are shown in red. IOP, intraocular pressure; m, months.
(DOCX)

**S2 Fig. Relationship between preoperative and 3-month postoperative IOP (mmHg).** The three preoperative IOP subgroups are indicated by color. On Pearson correlation coefficient analysis, r = −0.60 [−0.67; −0.51], p < 0.0001. M, month; IOP, intraocular pressure.
(DOCX)

**S3 Fig. Absolute IOP change (mmHg) over time in the nonglaucomatous and glaucomatous groups in previous studies that compared the two groups directly after cataract surgery.** Cataract surgery was conducted with phacoemulsification in all studies [14,21,27–29,31,34] except Shah et al., who used femtosecond-laser assisted cataract surgery [81]. The data of Coh et al. were not added because they only studied a single timepoint (4 months: nonglaucomatous eyes changed from 14.7 to 11.9 mmHg; POAG eyes changed from 14.9 to 12.2 mmHg) [41]. The data of Shingleton et al. (2006) were not added because they only studied two timepoints that were out of the x-axis range (nonglaucomatous eyes changed from 15.9 to 14.2 and 14.4 mmHg at 3 and 5 years; glaucomatous eyes changed from 18.4 to 17.0 and 16.6 mmHg at 3 and 5 years) [22]. Our data from nonglaucomatous patients were added to both graphs for comparison (red). d, day; GC, glaucoma; IOP, intraocular pressure; m, month; NG, nonglaucomatous; PACG, primary angle open glaucoma; POAG, primary open angle glaucoma; PXFGC, pseudoexfoliation glaucoma; w, week.
(DOCX)

**S4 Fig. IOP change in studies that controlled for RTM.** The details of the studies are summarized in S10 Table. (A) Two studies involved multiple preoperative IOP measurements and also had a control group that resembled the cataract-surgery group but did not undergo cataract surgery [6,84]. (B) Five studies involved multiple preoperative IOP measurements only [6,29,32,49,84]. (C) Five studies used the contralateral unoperated eye [46,47,83,91,92]. The legends show the glaucoma status of the eyes, and their preoperative IOP in brackets. CS, cataract surgery; GC, glaucoma; NG, nonglaucomatous; OHT, ocular hypertension, PACG, primary angle-closure glaucoma; preop, preoperative.
(DOCX)

**S1 Table. Previous studies examining the effect of phacoemulsification cataract surgery on IOP in nonglaucomatous eyes over time.**
(DOCX)

**S2 Table. Previous studies that searched for pre/perioperative factors that associate with greater IOP change in nonglaucomatous eyes.**
(DOCX)

**S3 Table. Previous studies on ocular macroanatomical measurements that associated with or predicted IOP drop after cataract surgery in nonglaucomatous or glaucomatous eyes.**
(DOCX)

**S4 Table. Univariable comparison of the 3-month whole cohort (*n* = 241) and the 12-month (*n* = 173) patients in terms of demographic, preoperative clinical, and operative characteristics.**
(DOCX)

**S5 Table. Preliminary multiple linear regression analysis of the ability of pre/perioperative variables to predict absolute IOP change (mmHg) and % IOP change at 3 months (*n* = 241).**
(DOCX)

**S6 Table. Preliminary multivariable analysis of the ability of pre/perioperative variables to predict IOP change (mmHg) in the 3-month cohort only and both the 3- and 12-month cohorts.**
(DOCX)

**S7 Table. Univariable analysis of the association between absolute IOP change (mmHg) at 3 months and pre/ perioperative variables in the 3-month whole cohort ($n=241$).**
(DOCX)

**S8 Table. Multiple linear regression analysis of the ability of LP and RLP to predict absolute IOP change at 3 months ($n=241$).**
(DOCX)

**S9 Table. Multiple linear regression analysis of the ability of pre/perioperative variables, including RLP, to predict absolute IOP change at 3 months in eye subgroups with preoperative IOP of 20–29, 15–19, or 10–14 mmHg (total $n=241$).**
(DOCX)

**S10 Table. Summary of studies that controlled for regression-to-the-mean when assessing the effect of cataract surgery on IOP.**
(DOCX)

**S1 Supplementary information.** STROBE Checklist and Protocol Percepolis-3.
(DOC)

## Author contributions

**Conceptualization:** Jean-Marc Perone.

**Data curation:** Jean-Marc Perone, Jean-Charles Vermion, Julie Francois, Alice Nesseler, Grace Gan.

**Formal analysis:** Christophe Goetz.

**Investigation:** Jean-Marc Perone, Jean-Charles Vermion, Julie Francois, Alice Nesseler, Grace Gan, Christophe Goetz.

**Methodology:** Jean-Marc Perone, Christophe Goetz.

**Project administration:** Jean-Marc Perone.

**Resources:** Jean-Marc Perone.

**Supervision:** Jean-Marc Perone.

**Validation:** Jean-Marc Perone.

**Visualization:** Yinka Zevering.

**Writing – original draft:** Yinka Zevering.

**Writing – review & editing:** Jean-Marc Perone, Jean-Charles Vermion, Yinka Zevering, Julie Francois, Alice Nesseler, Grace Gan, Christophe Goetz.

**Resources:** Jean-Marc Perone.

**Supervision:** Jean-Marc Perone.

**Validation:** Jean-Marc Perone.

**Visualization:** Yinka Zevering.

**Writing – original draft:** Yinka Zevering.

**Writing – review & editing:** Jean-Marc Perone, Jean-Charles Vermion, Yinka Zevering, Julie Francois, Alice Nesseler, Grace Gan, Christophe Goetz.

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
