## [Decision Letter · Decision Letter 0]

12 Oct 2025

PONE-D-25-40249Evolution of intraocular pressure after cataract surgery in nonglaucomatous patients: a post-hoc analysis of PERCEPOLIS clinical trial dataPLOS ONE

Dear Dr.  Perone,

Thank you for submitting your manuscript to PLOS ONE. After careful consideration, we feel that it has merit but does not fully meet PLOS ONE’s publication criteria as it currently stands. Therefore, we invite you to submit a revised version of the manuscript that addresses the points raised during the review process.

We look forward to receiving your revised manuscript.

Kind regards,

Academic Editor

PLOS ONE

Journal Requirements:

2. In the online submission form, you indicated that “The datasets generated during and/or analyzed during the current study are not publicly available according to French Law No. 2018-493 of June 20, 2018 on the protection of personal data (The General Data Protection Regulation (Regulation (EU) 2016/679) (GDPR: article 9) but are available from the Clinical Research Support Platform (Plateforme d’Appui à la Recherche Clinique [PARC]) of the Regional Central Hospital (CHR) of Metz-Thionville on reasonable request (email: projetrechercheclinique@chr-metz-thionville.fr, tel: +33 3 87 17 98 82). All non-archived data is subject to daily backups while all archived data is subject to duplicate storage at two different sites. This data processing is compliant with a baseline reference methodology (MR001) to which the CHR MetzThionville signed a compliance commitment on October 8, 2018.”

Reviewer's Responses to Questions

**Comments to the Author**

1. Is the manuscript technically sound, and do the data support the conclusions?

Reviewer #1: Yes

Reviewer #2: No

2. Has the statistical analysis been performed appropriately and rigorously? 

Reviewer #1: Yes

Reviewer #2: No

3. Have the authors made all data underlying the findings in their manuscript fully available?

Reviewer #1: Yes

Reviewer #2: Yes

4. Is the manuscript presented in an intelligible fashion and written in standard English?

Reviewer #1: Yes

Reviewer #2: No

5. Review Comments to the Author

Reviewer #1: As the statistical reviewer I will focus on methods and reporting. the paper is well-written and the limitations are discussed in detail by the authors.

Major

1) the study is now observational and needs to adhere to STROBE principles, please use an appropriate research checklist.

2) Analyses are generally appropriate but I would urge the authors to move away from statistical tests that only report p-values, in a cohort of limited size. Why aren't all relevant analyses conducted in a regression framework so association sizes and differences can be easily quantified, alongside their uncertainty in the form of 95% confidence intervals? these are more informative than p-values as well, and the focus should be on estimated effect sizes which should be interpreted cautiously and conservatively, considering the limited sample size (to answer this secondary research question, not the original question posed by the RCT).

3) what do the univariable (not univariate) analyses add? I'd recommend removal (unless justification is provided).

4) considering the levels of attrition I would expect to see sensitivity analyses using a multiple imputation framework as a mininum and even the exploration of best/worst case scenarios for the outcomes.

Minor

1) multivariable not multivariate (implies numerous outcomes)

2) please refrain from describing methods in the introduction, state the aims only, as you have done.

3) limited power is mentioned only for one subgroup analysis, it's arguably relevant for all analyses, please clarify that and rephrase to something like "power was limited and we would only be able to identify large associations with a high degree of certainty". I would also recommend removal of the outdated 10 subjects per covariate rule. this is so rough and depend on the type and distribution of the covariates, that is of very little use.

Reviewer #2: Abstract

• The abstract does not provide a clear and balanced overview of the study. It lacks precise reporting of the key outcomes (mean ± standard deviation) and does not fully reflect the depth of the analysis.

• The writing style is vague in places and does not capture the study's main strengths. A sharper, more structured abstract would enhance accessibility.

Introduction

• The introduction is generally well written, but the authors rely heavily on older references when discussing mechanisms of intraocular pressure reduction after cataract surgery.

• More recent studies using UBM and OCTA imaging should be cited, as they provide essential insights into angle parameters and trabecular meshwork morphology. Including such literature would significantly raise the relevance of the background section. DOI: 10.1097/IJG.0000000000001977, https://doi.org/10.1177/1120672119879331

DOI: 10.4103/meajo.MEAJO_20_19

• Overall, the reference list is outdated and should be updated with high-impact references.

Methods

• The surgical description is incomplete. While it is stated that one surgeon performed 88% of procedures, the level of experience of the other surgeons is not mentioned. Were they junior or senior surgeons? Was any comparison made across surgeons? This information is essential, as surgical experience may bias the outcomes. https://doi.org/10.1007/s10792-021-02103-6

• Given the large sample size, reporting data from the primary surgeon separately would be more rigorous, which could reduce statistical bias.

• The section on tonometry raises methodological concerns. The manuscript states that if the first two readings differed, a third was taken and the mean reported. However, it is unclear what proportion of patients required a third reading. A more robust approach would be three masked measurements for all participants, performed by an experienced ophthalmologist, with the mean reported.

• For each measurement device, the manufacturer, model, year, and country of origin should be specified in parentheses. This is standard practice and enhances reproducibility.

• More detail is needed regarding randomisation, masking, and handling of missing data.

Results

• The results section mixes data reporting with interpretation. For example, lines 219–220, 224–225, 229–230, 243–248, and 280–282 contain interpretative statements that should be moved to the Discussion.

• Reporting should follow a strict descriptive format (mean ± SD, p-values with exact numbers, and confidence intervals where appropriate).

• Tables and Figures:

o Table 1 appears redundant and could be removed, with its information integrated into the Introduction or Discussion.

o Graphs are visually complex and confusing. The authors should seek input from a statistician or biostatistician to simplify the visual presentation. Improved data visualisation (clearer legends, fewer but more focused figures) would strengthen the impact.

Discussion

• The discussion is overly long, repetitive, and at times speculative. Key results should be summarised more concisely, and interpretation should be better distinguished from the data.

• Mechanistic explanations (e.g., regression to the mean) are relevant but present with excessive certainty. The tone should be more cautious and evidence-based.

• Readers may find it challenging to identify the central message because of repeated statements and diffuse argumentation. A sharper, more structured discussion would be beneficial.

References

• The reference list contains a large number of outdated studies. Replacing these with recent (last 5–7 years) peer-reviewed articles would considerably enhance the manuscript’s credibility and relevance.

6. PLOS authors have the option to publish the peer review history of their article (what does this mean?). If published, this will include your full peer review and any attached files.

Reviewer #1: No

Reviewer #2: **Yes:** Farshid Karimi

---

## [Author Response · Author response to Decision Letter 1]

25 Nov 2025

Responses to Reviewers

Academic Editor:

Comment regarding our response to a comment from Reviewer 1

We have sought to address all reviewer comments to the best of our ability but could not address Comment #4 of Reviewer 1. This comment requests sensitivity analyses and exploration of best/worst case scenarios. Addressing this would require significant resources and we would have to request a long extension time to be able to complete this work. We feel that this effort and resource use is disproportionate for the type of work we are presenting. We would like to emphasize that Reviewer 1 has him/herself stated that “Analyses are generally appropriate” and that we have thoroughly addressed all other comments from both reviewers. We have also addressed all other comments of previous reviewers, including biostatisticians, with many new analyses (see Tables 2 and 3, S5 Table, S6 Table, S8 Table, S9 Table). We hope that our paper is now acceptable for publication in PLOS One.

Comment #1:

Reply: We have ensured that our manuscript is consistent with these style templates.

Comment #2:

2. In the online submission form, you indicated that “The datasets generated during and/or analyzed during the current study are not publicly available according to French Law No. 2018-493 of June 20, 2018 on the protection of personal data (The General Data Protection Regulation (Regulation (EU) 2016/679) (GDPR: article 9) but are available from the Clinical Research Support Platform (Plateforme d’Appui à la Recherche Clinique [PARC]) of the Regional Central Hospital (CHR) of Metz-Thionville on reasonable request (email: projetrechercheclinique@chr-metz-thionville.fr, tel: +33 3 87 17 98 82). All non-archived data is subject to daily backups while all archived data is subject to duplicate storage at two different sites. This data processing is compliant with a baseline reference methodology (MR001) to which the CHR MetzThionville signed a compliance commitment on October 8, 2018.”

Reply: French law is extremely strict about patient data protection: Law No. 2018-493 on the protection of personal data explicitly states that we cannot publicly share patient data, even when anonymized. Legally, we cannot add to or change our original Data Availability statement, which is:

“The datasets generated during and/or analyzed during the current study are not publicly available according to French Law No. 2018-493 of June 20, 2018 on the protection of personal data (The General Data Protection Regulation (Regulation (EU) 2016/679) (GDPR: article 9) but are available from the Clinical Research Support Platform (Plateforme d’Appui à la Recherche Clinique [PARC]) of the Regional Central Hospital (CHR) of Metz-Thionville on reasonable request (email: projetrechercheclinique@chr-metz-thionville.fr, tel: +33 3 87 17 98 82). All non-archived data is subject to daily backups while all archived data is subject to duplicate storage at two different sites. This data processing is compliant with a baseline reference methodology (MR001) to which the CHR MetzThionville signed a compliance commitment on October 8, 2018.”

Comment #3:

Reply: Reviewer 2 cited four papers. We did not feel that these papers pertained to our paper and therefore did not cite them.

Reviewer #1:

As the statistical reviewer I will focus on methods and reporting. the paper is well-written and the limitations are discussed in detail by the authors.

Comment: Thank you very much for your time and your pertinent and helpful comments on our paper.

Comment #1:

Major

1) the study is now observational and needs to adhere to STROBE principles, please use an appropriate research checklist.

Reply: We have provided a completed STROBE checklist for cohort analyses during resubmission.

Comment #2:

2) Analyses are generally appropriate but I would urge the authors to move away from statistical tests that only report p-values, in a cohort of limited size. Why aren't all relevant analyses conducted in a regression framework so association sizes and differences can be easily quantified, alongside their uncertainty in the form of 95% confidence intervals? these are more informative than p-values as well, and the focus should be on estimated effect sizes which should be interpreted cautiously and conservatively, considering the limited sample size (to answer this secondary research question, not the original question posed by the RCT).

Reply: To address this, we added effect sizes when discussing the linear mixed-modeling analysis (Lines 254-261) and the role of age (line 549-550). All other p-values in the paper were accompanied by parameters that allow the effect size to be evaluated, including partial r2 values in the regressions.

Comment #3:

3) what do the univariable (not univariate) analyses add? I'd recommend removal (unless justification is provided).

Reply: Our experience with other reviewers and readers of our manuscripts indicates that many physicians like to see univariable analyses, particularly those who are less well versed in statistics: many readily understand univariable analyses but are less able to interpret multivariable analyses. Nonetheless, to address this comment, we moved the univariable analysis table (previous Table 3) to the Supplementary Materials (now S7 Table).

Comment #4:

4) considering the levels of attrition I would expect to see sensitivity analyses using a multiple imputation framework as a mininum and even the exploration of best/worst case scenarios for the outcomes.

Reply: While we agree that this request is relevant, addressing it would require significant resources and we would have to request a long extension time to be able to complete this work. We feel that this effort and resource use may be disproportionate for the type of work we are presenting.

Comment #5:

Minor

1) multivariable not multivariate (implies numerous outcomes)

Reply: Thank you for this. We have made this change throughout the paper. The term “univariate” was also changed to “univariable” throughout.

Comment #6:

2) please refrain from describing methods in the introduction, state the aims only, as you have done.

Reply: We have deleted the term “multivariable” from the end of the Introduction.

Comment #7:

3) limited power is mentioned only for one subgroup analysis, it's arguably relevant for all analyses, please clarify that and rephrase to something like "power was limited and we would only be able to identify large associations with a high degree of certainty".

Reply: We rephrased the text as follows:

“Finally, the sample size was limited, meaning that we would only be able to identify large associations with a high degree of certainty.” (Lines 622-624)

Comment #7:

I would also recommend removal of the outdated 10 subjects per covariate rule. this is so rough and depend on the type and distribution of the covariates, that is of very little use.

Reply: We deleted this sentence.

Reviewer #2:

Comment: We would like to thank you very much for your time and your pertinent and helpful comments on our paper.

Abstract

Comment #1:

The abstract does not provide a clear and balanced overview of the study. It lacks precise reporting of the key outcomes (mean ± standard deviation) and does not fully reflect the depth of the analysis.

Reply: To address this, we expanded the Abstract to reflect other key findings and added the standard deviations for the mean IOPs as well as effect size estimates for independent variables.

Comment #2:

The writing style is vague in places and does not capture the study's main strengths. A sharper, more structured abstract would enhance accessibility.

Reply: To address this, the Abstract was edited so that it transmits the key messages of our study more clearly.

Introduction

Comment #3:

The introduction is generally well written, but the authors rely heavily on older references when discussing mechanisms of intraocular pressure reduction after cataract surgery.

Reply: To address this, we changed a key reference on the potential mechanisms driving PACG and POAG (ref 3, a 2014 review) to a more recent review (Wang et al. 2024 DOI: 10.2147/EB.S472920). Another reference on this topic (ref 36) was retained because it was published in 2020.

It should be noted that despite the abundant literature on the effect of phacoemulsification on IOP in nonglaucomatous eyes, and the factors that could drive this effect, a systematic review/meta-analysis on these issues has not yet been published. Therefore, we included some older references in our bibliography to demonstrate:

(i) The consistency over time and changes in clinical practice with which IOP has been found to drop after phacoemulsification in nonglaucomatous eyes (refs 13-35; summarized in S1 Table). These studies were also used to derive S1 Fig, which shows that the change in IOP in nonglaucomatous eyes observed in our study is similar to that reported by other studies.

(ii) The current state of the literature regarding factors that shape IOP after phacoemulsification (refs 13,14,29,32–35,37–41,15,42–51,17,52–58,18,20,23–25,28) (summarized in S2 Table). While many studies have searched for such factors, the vast majority were retrospective, involved small sample sizes, examined only a few variables, conducted univariable analyses only, and/or did not exclude the use of IOP-lowering medications after surgery. Including these studies in our Introduction highlights the contribution of our study to this field, since it involves multivariable analyses of prospective data obtained from a large randomized controlled trial on patients who were naïve for IOP-lowering medication both before and after phacoemulsification surgery. Many of the previous studies were also cited when discussing our findings in the context of the literature.

Comment #4:

More recent studies using UBM and OCTA imaging should be cited, as they provide essential insights into angle parameters and trabecular meshwork morphology. Including such literature would significantly raise the relevance of the background section. DOI: 10.1097/IJG.0000000000001977, https://doi.org/10.1177/1120672119879331

DOI: 10.4103/meajo.MEAJO_20_19

Reply: The three cited references were not directly relevant to our study because either they examined PACG/POAG patients (10.1097/IJG.0000000000001977 & 10.1177/1120672119879331) or phacoemulsification was not conducted (10.4103/meajo.MEAJO_20_19).

Notably, while reviewing these papers, we found two recent papers using LENSTAR biometry or Pentacam that assessed correlations between post-phaco change in IOP and preoperative ACD, lens thickness, change in ACD, change in iridocorneal angle, and change in anterior chamber volume (10.51329/mehdiophthal1467 & 10.3390/life15030381). These papers are relevant to S2 Table and thus were added to this table (refs 57 & 58) and cited in the Introduction and Discussion.

With regard to the point about UBM and OCTA, we added the following to the Study Limitations section:

“Third, it is possible that more precise measurements of ocular macroanatomical and especially microanatomical variables, such as phase-sensitive OCT measurements of the trabecular meshwork, could find associations between these variables and IOP change.” (Lines 593-595)

Comment #5:

Overall, the reference list is outdated and should be updated with high-impact references.

Reply: Please see our reply to Comment #3.

Methods

Comment #6:

The surgical description is incomplete. While it is stated that one surgeon performed 88% of procedures, the level of experience of the other surgeons is not mentioned. Were they junior or senior surgeons? Was any comparison made across surgeons? This information is essential, as surgical experience may bias the outcomes. https://doi.org/10.1007/s10792-021-02103-6

Given the large sample size, reporting data from the primary surgeon separately would be more rigorous, which could reduce statistical bias.

Reply: To address this, we sought to compare the ECL in the cases conducted by the main surgeron (JMP) to the ECL in the cases conducted by the other surgeons. However, we then realized that we had excluded the cases conducted by the other surgeons for the post-hoc analysis, exactly for the reason you stated: to limit confounding due to surgeon experience. To address this, we added the following texts to the Methods and Results sections:

Methods: “To minimize possible confounding due to differences in surgeon experience, we excluded eyes whose surgery had been conducted by surgeons other than the principal investigator (JMP).” (Lines 170-172)

Results: “Of the 292 patients included in the PERCEPOLIS trial, 22 (8%), 17 (6%), and 94 (32%) did not attend the 1-, 3-, and 12-month follow-up visit. Of the 270, 275, and 198 patients who attended these visits, 32, 34, and 25 were respectively excluded because IOP data were missing (n=3, 2, and 0, respectively) and/or surgery had not been conducted by the principal investigator (n=31, 32, and 25, respectively.” (Lines 221-225)

We also altered the figure showing patient distribution in the study (Fig 1) to reflect this change.

Comment #7:

The section on tonometry raises methodological concerns. The manuscript states that if the first two readings differed, a third was taken and the mean reported. However, it is unclear what proportion of patients required a third reading. A more robust approach would be three masked measurements for all participants, performed by an experienced ophthalmologist, with the mean reported.

Reply: This was the original methodology in the RCT protocol. In practice, we always take three readings and average them. To address this, we revised the Methods text as follows:

“All IOP measurements were conducted in the same way [64] with a slit lamp-calibrated Goldmann applanation tonometer (BQ 900; year 2012; Haag-Streit, Koniz, Switzerland). Specifically, at each study visit, IOP was measured three times in immediate succession between 10:00 and 12:00. The measurement average was then recorded.” (Lines 151-154)

Comment #8:

For each measurement device, the manufacturer, model, year, and country of origin should be specified in parentheses. This is standard practice and enhances reproducibility.

Reply: To address this, we added the manufacturer, model, year of purchase, and country of origin of the Goldmann applanation tonometer (Line 152), the IOL Master (Lines 157-158), and the phacoemulsifier (Lines 146-147) to the Methods section.

Comment #9:

More detail is needed regarding randomisation, masking, and handling of missing data.

Reply: The randomization and single-blinding procedures were described in detail in the original study (published in Cornea in 2021 doi:10.1097/ico.0000000000002822), as follows:

“In June 2015 to April 2016, a convenience series of patients was

---

## [Decision Letter · Decision Letter 1]

28 Apr 2026

Evolution of intraocular pressure after cataract surgery in nonglaucomatous patients: a post-hoc analysis of PERCEPOLIS clinical trial data

PONE-D-25-40249R1

Dear Dr. Perone,

We’re pleased to inform you that your manuscript has been judged scientifically suitable for publication and will be formally accepted for publication once it meets all outstanding technical requirements.

Kind regards,

Natasha Gautam, MBBS, MS

Academic Editor

PLOS One

Additional Editor Comments (optional):

The authors have appropriately addressed all the comments.

Natasha

Reviewers' comments:

Reviewer's Responses to Questions

**Comments to the Author**

1. If the authors have adequately addressed your comments raised in a previous round of review and you feel that this manuscript is now acceptable for publication, you may indicate that here to bypass the “Comments to the Author” section, enter your conflict of interest statement in the “Confidential to Editor” section, and submit your "Accept" recommendation.

Reviewer #1: All comments have been addressed

2. Is the manuscript technically sound, and do the data support the conclusions?

Reviewer #1: Yes

3. Has the statistical analysis been performed appropriately and rigorously? 

Reviewer #1: Yes

4. Have the authors made all data underlying the findings in their manuscript fully available?

Reviewer #1: Yes

5. Is the manuscript presented in an intelligible fashion and written in standard English?

Reviewer #1: Yes

6. Review Comments to the Author

Reviewer #1: I am satisfied with the authors' responses and the resulting changes to the paper. Nothing further to add.

7. PLOS authors have the option to publish the peer review history of their article (what does this mean?). If published, this will include your full peer review and any attached files.

Reviewer #1: No

---

## [Editor Report · Acceptance letter]

PONE-D-25-40249R1

PLOS One

Dear Dr. Perone,

I'm pleased to inform you that your manuscript has been deemed suitable for publication in PLOS One. Congratulations! Your manuscript is now being handed over to our production team.

Kind regards,

on behalf of

Dr. Natasha Gautam

Academic Editor

PLOS One